# BinaryFormer: 1-bit self-attention for long-range transformers in medical image segmentation and 3D diffusion models

**Mattias P. Heinrich**[1] (ID)                    MATTIAS.HEINRICH@UNI-LUEBECK.DE

[1] *Institute of Medical Informatics, University of Lübeck, Germany*

**Editors:** Accepted for publication at MIDL 2025

## Abstract

Vision transformers excel at capturing long-range interactions and have become essential for many medical image analysis tasks. Their computational cost, however, grows quadratically with sequence length - which is problematic for certain 3D problems, e.g. high-resolution diffusion models that require dozens of sampling steps. Flash attention addressed some limitations by optimising local memory access, but left the computational burden high. Quantising weights and activations for convolutions and fully binary networks are possible, but have to be trained at higher precision and often resulted in performance drops. For transformers recent studies have been limited to quantising weights in linear layers or exploiting the potential of sparsity in self-attention scores.

We present a novel scheme that not only enables a binary precision computation of the self-attention at inference time but also extends this to the training of transformers. To achieve differentiability we combine the bitwise Hamming distance with a learnable scalar query and key weighting. In theory this yields a 16-32$\times$ more resource-efficiency in arithmetic operations and memory bandwidth. We evaluate our model on three tasks with sequence lengths of N>1000: classification of images without patch-embedding, semantic 2D MRI segmentation and 3D high-resolution diffusion models for inpainting and synthesis. Our results demonstrate competitive performance and we provide an intuitive reasoning for the effectiveness of differentiable key-, query- weighting through Bernoulli sampling and distance interpolation.

https://github.com/mattiaspaul/BinaryFormer

**Keywords:** Long range, transformer, quantisation, diffusion, self-attention

## 1. Introduction

Vision transformer are a key building block for many modern medical deep learning architectures. Their strengths stems from the ability to learn both long-range interactions as well as localised features without the restrictions of the inductive bias of convolution operators. Yet, extending self-attention to sequences beyond a few hundred tokens becomes computationally complex often leading to hybrid architectures, in particular for 3D problems (Pang et al., 2023). The TransUNet (Chen et al., 2021) addresses the problem of quadratically growing complexity by using local convolutions at higher resolutions and only employing global self-attention within the low-resolution bottleneck. SwinTransformers (Cao et al., 2022) use transformer blocks throughout all resolutions of a U-Net for medical image segmentation but restrict the receptive field of the self-attention to a local shifted window.

| **Algorithm 1** Forward Attention | **Algorithm 2** Backward Attention |
|---|---|

```
1  def attention(x):
2      Q = x @ W_q
3      K = x @ W_k
4      V = x @ W_v
5      S = Q @ K.T / D**.5
6      P = softmax(S)
7      O = P @ V
8
9
10     return O
```

```
1  def attention_backward(dO):
2      dV = P.T @ dO
3      dP = dO @ V.T
4      dS = dsoftmax(dP)
5      dQ = dS @ K / D**.5
6      dK = dS.T @ Q / D**.5
7      dW_v = x.T @ dV
8      dW_q = x.T @ dQ
9      dW_k = x.T @ dK
10     return dW_v, dW_q, dW_k
```

Figure 1: Pseudo code for self-attention with potential channel reduction of values to $D_v$. Here, @ denotes matrix multiplication and T the transpose operation.

Besides image segmentation, diffusion models have been shown to benefit from transformers when formulation the denoising process within a vector-quantised (VQ) auto-encoder framework (Van Den Oord et al., 2017; Gu et al., 2022). Scaling VQ-diffusion to 3D medical images where the dimensions of the dimensions of latents should normally be $N > 10^3$ is challenging (Bieder et al., 2023). The complexity of a transformer becomes dominated by the sequence length $N$ when its number is larger than $4 \times D$ - the channel dimension. The ViT-S8 setting of DINO (Caron et al., 2021) with patch size of $8 \times 8$ and $N = 1600$ for images sized $320 \times 320$ and $D = 384$ is clearly bound in sequence length.

Next, we provide background on related work to limit memory and computational burden of long-range transformers. Our own concept focusses on the self-attention operation only and keeps the MLP, which is much less memory- and time-consuming in our $N >> D$ scenario, as is. The main operations of self-attention are the batched key-query matrix-product of $\mathbf{QK}^T$ followed by a softmax to produce the $B \times N \times N$ tensor $\mathbf{S_{QK^T}}$ which is matrix-multiplied with the value tensor $\mathbf{V}$ each requiring $\mathcal{O}(N^2D)$ multiply-accumulate (MAC) operations.

## 1.1. Related work

**Sparse attention approximation:** Driven by the long-sequences large language models, recent researched aimed to approximate attention matrix in linear time. (Hua et al., 2022) first remove the softmax to combine and rearrange the two matrix-multiplication within the attention block to a more efficient $\hat{V}_{lin} = Q(K^TV) \xrightarrow{\text{approx}} \hat{V}_{quad} = \text{Softmax}(QK^T)V$. This yields a complexity of only $ND^2$ but deteriorate the quality of the attention, which needs to be recovered by the proposed gated attention unit (GAU).

The Reformer (Kitaev et al., 2019) introduces locality-sensitive hashing (LSH). They share the linear mapping from the inputs to key and queries, equating $Q = K$. An approximate nearest neighbour search is implemented using LSH, restricting entries to only attend to other tokens from the same bucket. (Sun et al., 2021) aim to reduce the performance gap of such a drastic approximation - due to bucket imbalance issues - by learning a hash function for more efficient sparse attention. Both methods are evaluated on language benchmarks, but also on small images without patch embeddings (cf. Sec. B).Finally, (Yu

et al., 2022) evaluated a random attention matrix for vision transformers with surprisingly good results on image recognition tasks. They attribute the success of such an simplified architecture to power of the subsequent learned projection and MLP layers.

**IO-aware, memory efficient attention:** FlashAttention (Dao et al., 2022) point out that computational considerations for the trillions of MACs is not the only relevant aspect of throughput. The slow random access memory of the very large matrix $\mathbf{S_{QK^T}}$ makes it memory-bound in most implementations. It hence requires a tiled online matrix multiplication and softmax computation (Milakov and Gimelshein, 2018) to speed-up the process until it is computation bound again. FlashAttention is an exact computation (not approximate) that can reduce peak memory for most transformer models.

**Quantisation / Binarisation:** Much research has been dedicated to quantisation, ternarisation and binarisation-aware training of weights and activations in fully-connected and convolutional networks (Courbariaux et al., 2015; Rastegari et al., 2016). Either non-linear functions or linear weight coefficients are estimated to quantise tensors and create simulated low-precision integer values during the forward pass. For the backwards gradient computation, floating point arithmetic are used, but a large gain in inference speed and efficiency is obtained. The XNOR-net (Rastegari et al., 2016) reports speed-ups of $> 50\times$ on CPUs at the cost of an 18% drop in ImageNet accuracy. As an alternative post-training quantisation can be employed (Liu et al., 2021). A higher INT8 precision will often lead to insignificant performance reduction for large transformer models (Dettmers et al., 2022). In the medical domain MedQ (Zhang and Chung, 2021) and TernaryNet (Heinrich et al., 2018) were developed for low-bit quantisation of weights and activations in CNNs.

Many binarisation approaches (e.g. (Liu et al., 2022)) centralise weights by estimating a scaling factor that minimise the L2 error between real-valued and binarised weights before applying the sign function. BitNet (Wang et al., 2023) allows for a ternary activation and enables realtime inference of LLMs on the CPU. It however does not consider a more resource efficient training, but instead requires floating point precision for attention blocks and MLPs in the transformers.

To make the network trainable with quantised or binarised activations the mentioned works employ the straight-through estimator (STE) to enable backward propagation of a gradient through the otherwise non-differentiable sign function (see also Sec. 2.1). Crucially training has to be performed in memory-intensive floating point precision and so far only 8-bit quantisation has been successfully applied to reduce the complexity and precision of backpropagation. (Banner et al., 2018) propose a gradient bifurcation to enable 8-bit quantised backpropagation (QBP), whereas (Wiedemann et al., 2020) additionally sparsify the QBP for more efficient training. These promising steps lead to our research hypothesis that for the computationally most expensive part of long-range transformers, the self-attention computation, binarisation for training and inference could be viable.

## 1.2. Contributions and outline

Our method aims to provide first theoretical and empirical evidence for the performance of 1-bit self-attention training in the context of medical image analysis, with the following technical contributions:

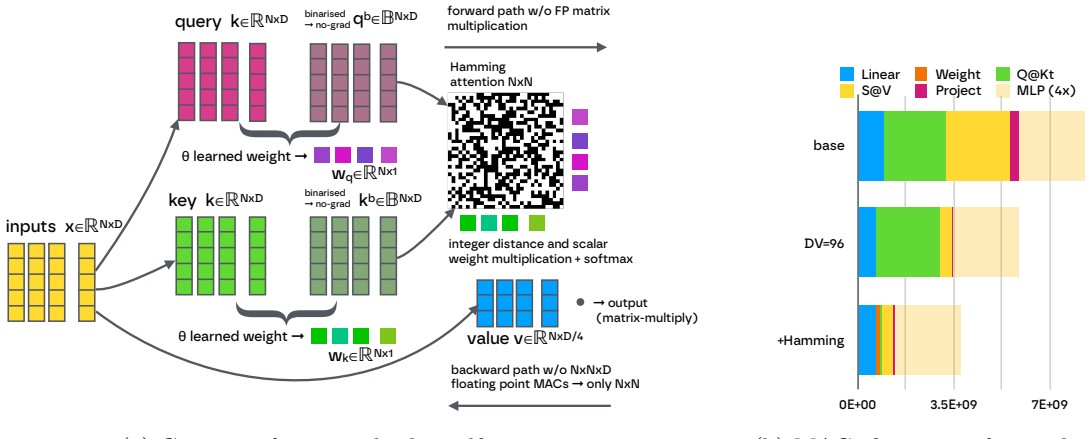

(a) Concept of proposed 1-bit self-attention
(b) MACs for 1 transformer layer

Figure 2: (a) Efficient 1-bit Hamming self-attention in long-range transformers using a splitting of weights and binarised values. (b) Reduction of MACs with $B = 6$, $N = 2048$, $D = 384$. Combining a 4x reduction of the value tensor with the proposed Hamming attention substantial reduces complexity, where now the MLP ($4\times$ channel expansion) dominates.

- a new derivation of differentiable binarised matrix-multiplication through the concept of bilinear interpolation of Hamming distances and Bernoulli sampling, and

- a further extension of the idea to a more efficient trainable scalar weighting among heads in multi-head attention.

In addition we provide empirical evidence across a range of practical experiments: 1) pixel-wise tokens for a proof-of-concept transformer architecture, which outperforms previous sparse-attention approaches on vision classification benchmarks; 2) fine-tuning of 2D foundation models for MRI segmentation, which demonstrates the advantages of the gradients derived from weighting scalars for 1-bit training, and 3) efficient 3D vector-quantised diffusion models that are on par with floating point precision transformers and can be trained and executed on low-power devices.

## 2. Method

Our method aims at reducing the number and precision of multiply-accumulate operations (MACs) in the self-attention step of transformers (see Fig. 2. It comprises four matrix multiplications: 1) Inputs $x$ are linearly transformed into $Q, K \in \mathbb{R}^{N \times D}$ and $V \in \mathbb{R}^{N \times D_v}$ by weight matrices $W_q, W_k \in \mathbb{R}^{D \times D}$ and $W_v \in \mathbb{R}^{D \times D_v}$. 2) Query $Q$ and key $K$ are matrix-multiplied together and subsequently scaled and softmax-normalised to obtain the attention weights $S$. 3) The attention scores are normalised and multiplied with the value $V$ to obtain the output $O \in \mathbb{R}^{N \times D_v}$. 4) The output is then linearly mapped by another linear weight matrix $W_p \in \mathbb{R}^{D_v \times D}$ to obtain the final output $y$ (see Fig. 1).

### 2.1. Binarisation-aware floating point training

First, we explore the simpler scenario to only speed up inference of transformers. The backward path through the self-attention layer may still comprise a large amount of floating point MACs but the method must be aware of the intended binarisation. One could simply use a sign function on key and query tensors as follows:

$$\text{sgn}(x) := (x \geq 0 \rightarrow +1) \wedge (x < 0 \rightarrow -1), \tag{1}$$

yielding $\mathbf{Q^b} = \text{sgn}(\mathbf{Q})$ and $\mathbf{K^b} = \text{sgn}(\mathbf{K})$. But this would completely stop the backward gradient flow to those linear layers. This does, however, not necessarily lead to an untrainable transformer. As pointed out amongst other by (Yu et al., 2022), a randomly initialised and frozen attention matrix leads to a mere drop of 1.4% ImageNet-1K accuracy compared to other approximations. We employ the **sign stop grad** function as a baseline. Here, $\mathbf{QK}^T$ is efficiently computed using the Hamming distance of the two binary operands:

$$\mathbf{Q^b}_i \mathbf{K^b}_j = c - 2\Xi\{\mathbf{Q^b}_i \oplus \mathbf{K^b}_j\} \tag{2}$$

where $\oplus$ defines an XOR operator and $\Xi$ a bit-count over the $c$ bits in the rows of $\mathbf{Q^b}$ and $\mathbf{K^b}$. Compared to single or half precision floating points the memory access will be reduced by $32\times$ or $16\times$ respectively. The energy consumption of INT8 MACs is $6\times$ lower as compared to FP16 (Zhang et al., 2022), with a substantial further reduction for INT1.

Next, we turn to binarisation-aware training using floating point arithmetic for back-propagation to alleviate the break in gradient flow from before. In (Hubara et al., 2016) and (Wang et al., 2023) an adhoc **step-through estimator** is proposed using:

$$\partial \text{sgn}/\partial x \approx (|x| \leq 1 \rightarrow 1) \wedge (|x| > 1 \rightarrow 0) \tag{3}$$

This simple approximation leads to surprisingly good results and enables good convergence of the network weights during training, but revisiting Algorithm 1 lines 5 and 6 reveals that the derivatives with respect to $\mathbf{Q^b}$ and $\mathbf{K^b}$ will comprise the floating point operand `dS` for matrix multiplications in the backward pass, hence no computational gain is achieved. To our knowledge all current binary networks are only beneficial for inference.

### 2.2. Differentiable Weighing of Hamming distances

The key contribution of this work is to introduce a differentiable weighing of the binarised matrix multiplication to alleviate the aforementioned limitations and improve energy efficiency of **training** long-range transformers as well. Similar to (Courbariaux et al., 2015) we can make use of a stochastic version of the binarisation in Eq. (1) using a Bernoulli distribution sampling with probabilities defined as $\rho(x) = \max(0, \min(1, \frac{x+1}{2}))$ to yield a probabilistic sign function: $\text{psgn}(x) := (\text{with prob. } p = \rho(x) \rightarrow +1) \wedge (\text{with prob. } 1 - p \rightarrow -1)$. This function is commonly applied after an activation $\sigma$ that reduces the range of input values $x_0$. We use a modified hyperbolic tangent in our experiments - specifically

$$x = \sigma(x_0) = 2\frac{\tanh(\beta x_0)}{\min(\beta, .75)} \text{ , with } \beta = 0.5. \tag{4}$$

The Bernoulli sampling on its own does not enable a backpropagation of the loss, we hence propose a bilinear weighting of two samples. When sampling $x \in \mathbb{R}^{N \times D}$ twice to obtain $y^{(1)}$, $y^{(2)} = \mathrm{psgn}(x)$ we can derive weights based on the binarisation distances $d^{(1)} = \frac{1}{D} \sum_j^D |x_j - y_j^{(1)}|$ and $d^{(2)} = \frac{1}{D} \sum_j^D |x_j - y_j^{(2)}|$. Note that $d \in \mathbb{R}^N$ is a scalar value for each token. We then define $w^{(1)} := \frac{d^{(2)}}{d^{(1)} + d^{(2)}}$, $w^{(2)} := \frac{d^{(1)}}{d^{(1)} + d^{(2)}}$ and $y^* = w^{(1)} y^{(1)} + w^{(2)} y^{(2)}$ using the differentiable bilinear weights. Now, the gradients are indirectly derived for the input values $x$ without the need for the adhoc STE.

Within the context of self-attention the procedure is performed for both queries and keys, leading to a sum of four operands. Crucially, the linear weighting can be computed as scalar pointwise product after computing the respective Hamming distance $H_{QK} = \Xi\{\mathbf{Q^b}_i \oplus \mathbf{K^b}_j\}$ or binary inner-product $QK^{(b)} = c - 2H_{QK}$, keeping the computational and memory efficiency of the forward path. Since no derivative for $\mathbf{Q^b}, \mathbf{K^b}$ but only with respect to the weights $w_q, w_k$ we can rewrite the backward path and obtain $\partial w_q, \partial w_k$ without any floating point matrix multiplication.

Computing four Hamming distances partially diminishes the computational gains. We therefore extend our concept with a trainable unary weighting per token (see Fig. 2, right). Instead of explicitly defining the binarisation error distance, we concatenate $x$ and the value $z = \mathrm{sgn}(x)$ obtained by binarisation with the non-differentiable, non-probabilistic sign function. We learn a weight $w = \theta(x||z)$ by feeding them into a small shared MLP $\theta$ with trainable parameters, separately for keys and queries. For backpropagation we redefine $\mathbf{S} = \mathbf{QK^{(b)}} \cdot (w_q \odot w_k^T)$ in line 5 of Fig. 1 (left), where $\odot$ evaluates a pointwise outer-product of the broadcasted scalar weights. Given $\partial \mathbf{S}$ from line 4 in Fig. 1 (right) we get $\partial w_q = \sum_j^N ((\partial S \cdot \mathbf{QK^{(b)}}) \odot w_k^T)_j$ which are $N \times N$ pointwise multiplies followed by a reduction along the last axis and $\partial w_k = \sum_j^N ((\partial S \cdot QK^{(b)}) \odot w_q)_j$ reduced along the second last axis.

## 3. Experiments and Results

Next, we summarise our experimental setting (details are found in Sec. A and https://github.com/mattiaspaul/BinaryFormer) before presenting the results for different variants of the self-attention mechanism. Pixel-wise long-range transformers experiments compared to sparse attention (e.g. (Sun et al., 2021)) are found in Sec. B.

### 3.1. Finetuning DINO ViT-S8 for semantic MR segmentation

We consider a setting in which vision transformers with small patches ($8 \times 8$) is trained together with a convolutional decoder to predict semantic segmentations for three different open-source MRI datasets: AMOS (abdominal organs, 7 labels) (Ji et al., 2022), Spine (vertebrae and discs, 19 labels) (van der Graaf et al., 2024) and CrossModa (vestibular schwannoma, 1 label) (Dorent et al., 2023). Details of the modified 2D models with 19M and 8M parameters in transformer encoder and decoder respectively are specified in Sec. A.1. We reinitialised all weights in the attention blocks - approx. 23% of transformer weights - and in addition all MLPs of the eight last blocks.

The results for the baseline and three variants, including our proposed differentiable Hamming attention, are listed in Tab. 1 and visualised in Fig. 3. Our fully-binarised ap-

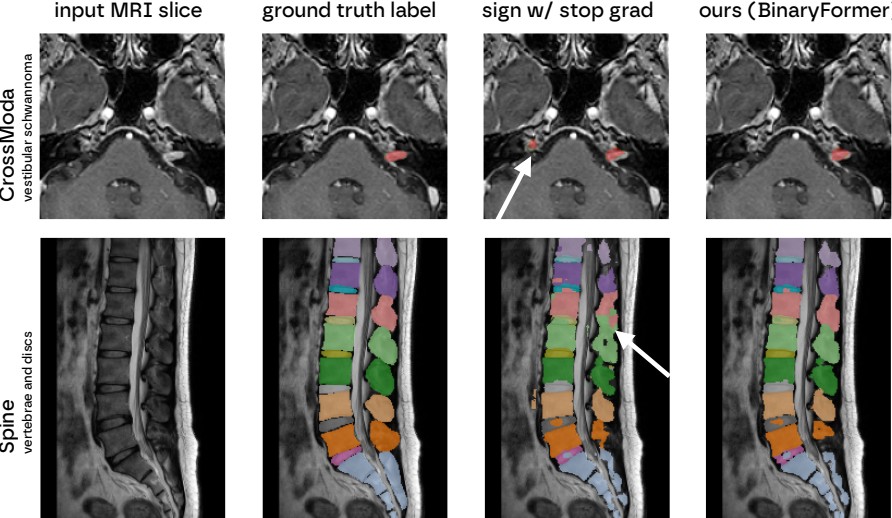

Figure 3: Qualitative results comparing binary self-attention with stop grad and our BinaryFormer that is trained with 1-bit precision. Arrows point to improvements.

| Model | base | sign | sign st.gr. | Hamming |
|---|---|---|---|---|
| CrossModa | $69.9\% \pm 21.7\%$ | $65.1\% \pm 23.5\%$ | $61.4\% \pm 22.9\%$ | $71.4\% \pm 23.0\%$ |
| AMOS | $64.2\% \pm 24.1\%$ | $62.9\% \pm 25.9\%$ | $52.4\% \pm 32.3\%$ | $64.1\% \pm 25.9\%$ |
| Spine | $60.3\% \pm 26.2\%$ | $57.1\% \pm 26.8\%$ | $58.6\% \pm 25.5\%$ | $60.5\% \pm 25.8\%$ |
| Average | **64.78%** | **61.68%** | **57.47%** | **65.33%** |

Table 1: Dice scores for different models evaluated on three MRI datasets

proach performs on par with the floating point baseline across all tasks. It even outperforms the **sign** variant with STE, which may only improve efficiency for inference but not training. The naive stop-grad variant clearly falls short highlighting the importance of sufficient gradient back-flow to the query and key weight matrices. We also compared three variants for the differentiable weighing for the CrossModa dataset - directly computed bilinear weights with four Hamming distance, trainable weights with or without grouped linear MLPs - and found only negligible differences of $< 0.5\%$. Note that the ViT is not ideal for segmentation - being single-scale - and e.g. (Xie et al., 2021) could lead to better results.

### 3.2. 3D VQ Diffusion and Inpainting

Finally, vector-quantised diffusion models with absorbing transformers (Bond-Taylor et al., 2022) are trained as detailed with the methodological specifics in Sec. A.2. We evaluated the ability of our long-range transformers to inpaint or synthetise 3D medical volumes with high efficiency. The OASIS 3D brain dataset is preprocessed as in (Hering et al., 2022) with a resolution of $160 \times 224 \times 192$ voxels. The convolutional decoder is only applied to patches of size $32^3$ and comprises 12M parameters. The transformer uses a positional embedding for a quantised latent space of shape $10 \times 14 \times 12$ ($N = 1680$), 8 blocks and the

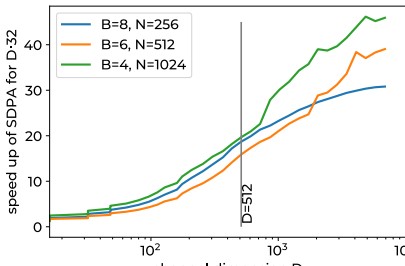

| Model | PSNR | SSIM |
|---|---|---|
| base | 26.09 | 0.944 |
| sign | 26.54 | 0.946 |
| sign st.gr. | 26.06 | 0.944 |
| Hamming | 26.58 | 0.946 |
| random | 17.65 | 0.697 |

Figure 4: Left: Computational speed experiments for decreasing channels by $32\times$ as possible with the proposed binary attention. Right: Quantitative evaluation of diffusion models for inpainting on 3D brain MRI.

same hyperparameters (e.g. $D = 384$) as before without any pre-training. The codebook used EMA and $8 \times 32$ dimensions with 64 dictionary entries each. Posterior (codebook) loss, L1 reconstruction error and perceptual dissimilarity were all equally weighted.

For inference we start from the (partially) randomised token masks (centre region unknown) and sample/predict one new mask entry at a time for 200 steps with a temperature of 0.8, which takes less than 1 sec. on GPU. Quantitative inpainting results with severely masked unseen images that are compared to the ground truth reconstructions with PSNR and SSIM are in Tab. 4. We also include the option *random*, which only uses the VQ-AE without any denoising steps. All models perform comparably well - greatly improving over the initial guess - even the stop-grad version of the sign-activated attention. Visual mask reconstruction results and synthetically sampled 3D volumes can be found in Sec. C.

**Computational speed:** A detailed analysis of runtimes on different computational architectures is not considered here - as it depends greatly on implementation choices and low-level code engineering (cf. (Dao et al., 2022)). We experimentally confirmed the increased throughput of approx. 32-fold compared to FP32 using 1-bit cutlass matrix-multiplications. In Fig. 4 we provide measurements of the scaled-dot-product-attention in pytorch on an Ampere GPU for several settings with respect to the speed-up of dividing the channels by this factor of 32. Around $D = 512$, an improvement of $\approx 15\times$ could be achievable with binary matrix multiplication within the attention block. Further tests are found in the Sec. D.

## 4. Discussion and Conclusion

This work demonstrated, as a first, the feasibility of fully 1-bit differentiable attention computation in forward and backward pass for long-range transformers in the context of medical imaging. When only binarisation-aware training for faster inference is required the step through estimator provides sufficient quality. Yet to train a transformer with binary precision for efficiency gains the proposed new Hamming attention works best and substantially improve over the naive stop-grad variant.

More experiments on different models could yield further insights on the relevance of precision in the self-attention step and analyse the impact of Bernoulli sampling compared to other binarisations during training. An interesting alternative derivation of the feasibility of

backpropagating through the query and key weights would be the use of trainable attention biases (Press et al., 2021) combined with non-differentiable scores. Future work could also combine our binarisation for attention with other parts of the transformer - the MLP and linear weight matrices, cf. (Wang et al., 2023).

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

## Appendix A. Additional details of the experimental settings for the different methods

### A.1. Finetuning of foundation models

It is usually challenging to train powerful transformers from scratch on a small (medical) dataset. We hence explore the use of the DINO pre-trained transformers (Caron et al., 2021). Specifically, we employ the ViT-Small variant with 12 blocks, a patch-size of $8\times8$ and a feature dimension of $D = 384$ split into six heads for self-attention. The small patch size leads to a large number of tokens, e.g. when using images of size $320\times320$, $N = 1600$ making the learning of long-range interactions particularly important. To evaluate the different concepts of binarised attention against floating point arithmetic, we modify the DINO model as follows: 1) add a new convolutional decoder as segmentation head (see details below); 2) reduce the channel size for the value tensor from 64 to 16 (reduces FLOPS by 50%); 3) keep the initialisation of patch-embedding and the MLPs of blocks 0-3, 4) reinitialise the weights in all attention blocks, and 5) also reinitialise the the MLPs of block 4-12. This comes closer to training from scratch and enables us to explore the ability of the binarised attention methods to back-propagate into earlier blocks. But it will limit the benefit of pretraining and only maintain some generalisable knowledge of low-level image understanding from the MLPs of the first four blocks.

The convolutional decoder is design as follows: to increase the resolution by $8\times$ we employ three blocks with transposed convolutions with stride= 2. The input channel dimension of 384 is expanded to 1024 and halved at each resolution. Each decoder block comprises pixel-shuffle and pixel-unshuffle blocks with a residual $1\times1$ convolution part to mainly restrict the learning on the localised patches. This avoids moving too much of the complexity from the transformer to the convolutional part. For the last resolution only, we use two standard $3\times3$ convolutions to reduce block artefacts.

All models are trained for 12000 iterations with a batch size of 12 and evaluated on hold-out validation patients, see Sec. 3.1. A linear ramp-up and step decay is used as learning rate scheduling for Adam. All datasets are open source and freely accessible with CC license and where accessed through the FMV-CVPR24 workshop https://fmv-cvpr24workshop.github.io.

### A.2. Long-range transformers for diffusion

To fully unleash the potential of the binarised long-range attention, we also consider 3D vector quantised diffusion models. As pointed out by (Bieder et al., 2023) and (Friedrich et al., 2024) 3D denoising diffusion models have lavish memory usage that make them tricky to train in 3D. At inference they also consume a lot of energy since dozens of steps (each a 3D model) need to be evaluated. A promising alternative to U-Net based DDPMs (Wolleb et al., 2022) are vector quantised (or latent) diffusion models ((Gu et al., 2022)) that

comprise 1) a vector-quantised convolutional autoencoder (VQ-AE) which only provides a reconstruction loss and sparse codebook embedding (see also (Esser et al., 2021), and 2) a transformer that needs to learn long-range interactions to reverse the masking/denoising of quantised latent codes. These parts are trained separately.

We again aim to minimise the influence of the convolutional part to focus on pure transformer performance. For a given 3D scan with $160 \times 224 \times 192$ voxels we subdivide the image space in non overlapping patches of $32^3$ voxels. The VQ-AE is hence not able to directly learn long-range interactions - with the potential downside of block artefacts for reconstructed whole images. The VQ-AE was trained on 250 scans, yielding 52500 3D patches for 8000 iterations with a batch-size of 64.

We employed the absorbing transformer concept with parallel token prediction in training to efficiently train the denoising sampler (Bond-Taylor et al., 2022) for 21000 iterations with a batch-size of 12 and 256 time-steps. The latent space of the transformer is later used on the whole domain and set to comprise up to $10 \times 14 \times 12$, $N = 1680$ tokens. Each $32^3$ patch of the auto-encoder is represented as $2 \times 2 \times 2$ latent code. An exponential moving average codebook is learned with a feature dimension of 384. In addition to a standard reconstruction and codebook loss, the VGG perceptual loss is employed on three orthogonal slices of the 3D output patches.

We make one important adjustment in our architecture to compensate for the weaker VQ-AE (with separate patches). The codebook is split into 8 sub-codebooks each comprising just 64 entries (i.e. 6 bits of information). That way the final layer output is treated as a multi-head prediction and the loss function is the average of 8 cross-entropy scores. This immensely improved training convergence and visual quality of the reconstructions.

## Appendix B. Pixel-wise transformer

We also analyse setting, used e.g. in (Sun et al., 2021), where all pixels of small images (CIFAR10, MNIST) are directly considered as input to a transformer encoder with eight blocks without any patch-embedding. Using RGB and two spatial coordinates as input leads to a five-channel input and the sequence lengths is $N = 1024$. We set the channels to $D = 384$ and use $H = 4$ heads. The intermediate MLP expands the channels four-fold, whereas the value dimension is always reduced to $D_v = 64$. The models are trained for 3000 iterations with a batch-size of 32. Sparse attention with learned hashing (Sun et al., 2021) reached 45.2% accuracy and Reformer (Kitaev et al., 2019) only 38.1% which falls short of the vanilla self-attention transformer (Vaswani et al., 2017). These results highlight the difficulty of the task (which is considered nearly solved when using image patches). Our improvements over those approaches are highlighted in Tab. B.

| Model | base | sign (STE) | sign st.gr. | Hamming | Reformer | Learn. Hash. |
|---|---|---|---|---|---|---|
| CIFAR10 | 51.0% | 51.7% | 47.2% | 49.1% | 38.1% | 45.2% |
| MNIST | 95.4% | 96.9% | 91.5% | 93.9% | - | - |

Table 2: Accuracy for different models on pixel-wise CIFAR10 and MNIST classification.

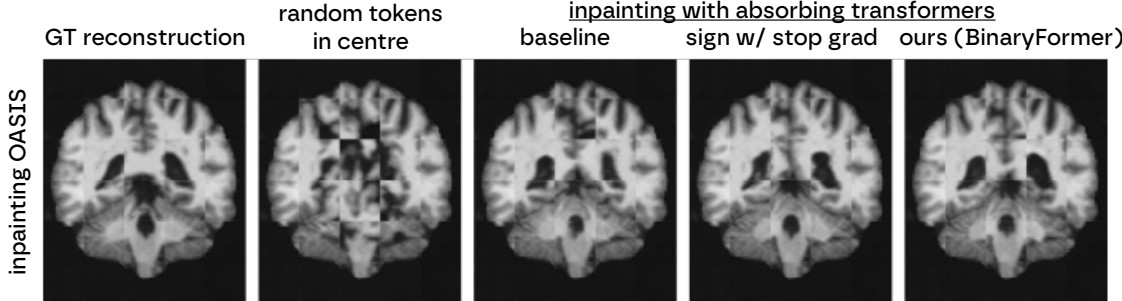

Figure 5: Visual comparison of inpainting using variants of transformers for the denoising vector-quantised diffusion model.

### B.1. Classification results

While image- or patch-based classification of CIFAR10 and MNIST is an easy problem, recent work on long-range sparse attention (Sun et al., 2021) highlighted the challenges for pixel-wise transformers. Tab. B demonstrates that all our considered variants outperform two state-of-the-art approaches to efficient hashing, Reformer and Learned Hashing Attention (LHA). The binary forward path using the sign activation together with the step through estimator for backward computation performed best, followed by the floating point precision baseline. Our differentiable Hamming attention, which can be trained with 1-bit precision for key-query computations during both paths is approx. 2% points better than the other alternative of using sign activations coupled with no gradient (equivalent to a random attention matrix).

## Appendix C. Visual 3D diffusion model results

Following the experiments described in Sec. 3.2 and Sec. A.2 we present visual examples from coronal slices of the reconstructed or synthesised volumes. As mentioned in the numerical results, all methods perform comparably for this task - enabling a plausible inpainting of the missing information. Synthetic samples vary in their quality but look generally promising. Note, that we expect substantial improvements by using a stronger convolutional VQ-AE but left this out purposefully to focus on transformers. As can be seen in the next section, the limitation to a channel width of $D = 384$ is likely unnecessary as the throughput comes much closer to the compute limit when increasing this hyper-parameter for BinaryFormers.

## Appendix D. Measured computational speed

Implementing low-precision network computations is not always straightforward. Pytorch only provides experimental support for INT8 matrix multiply, however without directly supporting batches. Lower precision integers are not yet integrated at all. Starting with the Ampere architecture Nvidia supports INT4 tensors algebra, including binary INT1 operations (defined in instruction set sm80). The former provide a very high performance 4992 TOPS for pure binary large-matrix multiplication compared to the 19.5 TFLOPS

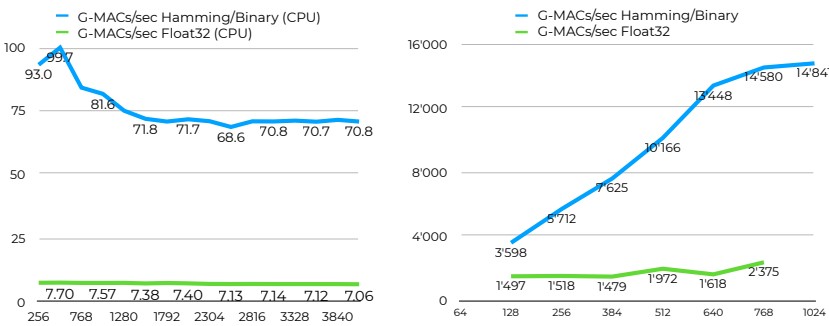

Figure 6: Throughput in G-MACs/sec Left: For `np.binary_count` vs `np.matmul` on CPU with $B = 1$, fixed channel dimension $D = 32$ and increasing sequence length $N$. Right: For MPSGraph `HammingDistance` vs. `matrixMultiplication` for $B = 16$, fixed $N = 8192$ and increasing $D$.

(FP32) of the same card (256x speed-up) and 312 TFLOPS (bfloat16) which still yields the expected 16x speed-up. Yet the kernels are not yet exposed outside C++/Cutlass and difficult to integrate in pytorch.

Therefore, we explored two benchmarks that are in particular interesting on low-power devices, here an M2 Max. Fig. 6 compares binary Hamming distances and floating point matrix multiplication with numpy's `np.binary_count` (available from version 2.2) and `np.matmul`. A fairly constant speed-up of 8-9× is achieved using the INT1 computation. Next, we implemented a custom MPSGraph kernel for integrating the latest `HammingDistance` instruction for GPUs into pytorch and evaluated a full self-attention block with fixed sequence and increasing channel depth. Here, a slight ramp-up, that is due to memory constraints and overhead for calling the custom functions, can be seen for the binary computation. It then reaches ≈ 15 TOPS and improves over the floating point precision by about 6×. Note that the aforementioned Nvidia numbers are for matrix multiplications only and SDPA oftentimes cannot reach more than 25-50% of the theoretically achievable performance. Details can be found in our github repository.

