# OpenReview forum: "BinaryFormer: 1-bit self-attention for long-range transformers in medical image segmentation and 3D diffusion models"
_MIDL.io/2025/Conference — MIDL 2025 Oral_

### Official Review · Reviewer_5rRZ · 2025-02-17

**Confidence:** 4
**Preliminary Rating:** 4
**Recommendation:** Poster
**Final Rating:** 5

**Summary:**

Self attention is an important building block in modern network architectures. It can process relationships within e.g. tokens in a sequence or patches in an image. Its main limitation is that it scales quadratically in the sequence length, but it is usually desirable to extend this length as much as possible. This work introduces a novel binarized formulation of self attention, which is a lot more efficient in terms of compute and memory, not only in inference but also during training.

**Strengths:**

The proposed method is well motivated and and provides a novel way to tackle the scaling-problem of self attention for long sequences. In multiple experiments the author shows that with respect to performance this novel formulation is on par with the traditional floating-point formulation, while consuming a lot less resources. This is shown for a baselines as well as the different components of the proposed method. This method appears to be especially relevant for medical image processing due to the high dimensionality of the data.

One of the core ideas of the paper is replacing the expensive QK-product with a binarized version. Instead of a scalar product, each entry is therefore replaced with the Hamming distance between the two factors, which acts like the natural equivalent for the binary product. This alone is however not differentiable, but the authors then also first show a more primitive way and then their proposed method of approximating gradients of this non-differentiable operation, which means that this can be used as a direct replacement of the standard self attention block and optimized with gradient based optimizers.

The experiments include segmentation and inpainting tasks which are challenging and and show the viability of the proposed method.

**Weaknesses:**

The segmentation experiments only include 2D models. It would have been interesting to include some high resolution 3D task to prove how the application to really show the benefits of the reduced resource consumption, especially in the medical context. I understand that this might be a more involved experiment.

In general, the writing could be a little bit clearer, including some notations that could be improved. I elaborated some of the issues I came across in the detailed comments. However, I think they can be addressed.

**Detailed Comments:**

There are some details and suspected typos that I would like to ask the author to check for a final version:

- In Equation 3 in the second clause I assume the $\leq 0$ should not be there.

- The equations are referenced as e.g. (2.1) in Section 2.2, but the equation is just labeled as (1) which confused me first. (Same for some Figures), I’d check that the labeling and References are consistent - but I’m not sure if this is part of the template.

- In Algorithm 2 of Figure 1, some line breaks seem to be missing. They would enhance readability.

- In Section 2.2. the formula for $\partial w_k$ and $\partial w_q$ the index $j$ of the sum is not being used. Furthermore, I believe that just above, in the same paragraph, the in the redefinition of the $S$, the operators $\cdot$ and $\odot$ are swapped, and/or the precedence is unclear. Maybe some parenthesis would help. I did not redo the computation, and while I can follow the general idea, I’d appreciate a clearer notation.

- In Appendix B a non-existent Table is being referenced.

- Appendix C mentions images of some coronal slices that are not included.

- On page 14 the plots from Fig. 6 appear without any caption. (Maybe related to the point above.)

**Justification Of The Final Rating:**

The authors addressed the open questions appropriately, and updated the notation in the manuscript to make it more easily readable and understandable. Considering the very interesting contribution of the paper, I am positive that the proposed method of this much less resource-hungry attention block can have a profound impact in various applications.

**Justification Of The Preliminary Rating:**

The paper presents a novel method to tackle the problem of the expensive self attention for long input sequences. The method is composed of multiple components that work in tandem, and allow a reduction of resource consumption in terms of memory, operations and also energy, while not compromising on performance. All these advances are highly relevant for medical image processing, but can also be beneficial outside of this domain.

**Questions To Address In The Rebuttal:**

1. In section 2.2. the scaled tanh-activation is introduced, I’m wondering whether this is actually a crucial part, or whether other activations would work just as well. Can you motivate its use and the significance of the choice of $\beta = 0.5$? I’m mainly asking because this formula was so prominently featured in the manuscript.

2. Regarding the baseline “sign stop gradient” version: Is it correct that the gradient only flows through the V values? If so, that means that the Q and K are computed using fixed matrices $W_q, W_k$. Do you have any intuition on what influence the initialization of these two matrices has in this case?

3. The binarization distances $d^{(i)}$ are defined as $l^1$ norms. Is there a motivation for this choice or could other choices also be viable?

**Special Issue:**

Yes

---

> ### Author Response · Authors · 2025-03-07
>
> We are very grateful for the very detailed review and excellent remarks in particular the appreciation that "All these advances are highly relevant for medical image processing, but can also be beneficial outside of this domain."
>
> Q: Suspected typos, missing brackets in equation and line breaks in pseudo-code
>
> A: The comments were very helpful and we have amended all suggested details. We fixed the incorrect Latex references and added the correct figure of the synthesised inpainting (diffusion model) results in the appendix.
>
> Q: Relevance/prominence of $\beta$ in scaled tanh-activation
>
> A: This is indeed an interesting question. We ran some additional experiments with a standard hyperbolic tangent and found only a small deterioration of on average 1.3% Dice for the 2D segmentation models. Hence we would tend to argue that it is not entirely crucial.
>
> Q:  Inclusion of some high resolution 3D task
>
> A: We appreciate the suggestion for performing additional experiments for 3D segmentation and will address this in future work. One initial reasoning to focus on 2D was the possibility to leverage pre-trained transformer models and evaluate the transferability. However, this did not turn out to be crucial and hence training a 3D model from scratch is a natural next step that could further demonstrate the gains in memory/runtime efficiency of binary attention.
> As a small note: the VQ diffusion model for the inpainting task was indeed trained for 3D and we experimented  latent space dimensions of up to 16x24x20 hence a  sequence length of 7’680 (and 59 million binary attention scores for each sample) with success.
>
> Q: Influence of initialisation of untrained query / key matrices when using stop-grad sign
>
> A: This is an excellent comment - and yes there would be some influence of the random initialisation. As mentioned in prior work (W Yu MetaFormer) random attention matrices can be useful. To my understanding this is likely because multiple spatial arrangements are weighted (with still optimisable parameters) in the projection part from different heads after the attention mechanism. But further analyses would be necessary whether they e.g. lose effectiveness of capturing positional information.
>
> Q: Choice of $\ell_1$ for binarisation distances
>
> A: We did not experiment with squared norms yet but would expect them to work similarly well.
>
>
> Additional comment: we recreated the concept figure Fig. 1 as suggested by other reviewers and hope this is now also a bit clearer.

---

> > ### Comment · Reviewer_5rRZ · 2025-03-11
> >
> > I'd like to thank the author for thoroughly addressing the questions and updating the manuscript accordingly. The updated notation in section 2 now is much clearer in my opinion. I also appreciate the updated figure 2 which is also more readable, and it is nice to see the visual examples in figure 5.

---

### Official Review · Reviewer_wjXn · 2025-02-20

**Confidence:** 3
**Preliminary Rating:** 3
**Recommendation:** Poster
**Final Rating:** 4

**Summary:**

This paper present a novel scheme that enables a binary precision computation in training and inferencing time.

**Strengths:**

The authors proposed a 1-bit self-attention for transformer and diffusion-based models. They combine the bit-wise Hamming distance with a learnable scalar query and key weighting. The idea of this paper is interesting.

**Weaknesses:**

Figure 2 is unclear and does not effectively convey the main point; the logical flow could be improved. Additionally, there are no quantitative or qualitative comparisons with other methods to demonstrate the effectiveness of the proposed approach. The experimental sample sizes are also small.

**Detailed Comments:**

Please refer to the weakness section.

**Justification Of The Final Rating:**

The authors addressed my concerns well, and they also redraw some diagrams to help the readers understand the method better. Overall, I think this is an interesting paper, and I've updated my final rating accordingly.

**Justification Of The Preliminary Rating:**

Overall, this is a good paper that proposes an interesting framework leveraging 1-bit self-attention. However, the experiments are not thorough, and some essential tests are missing. I recommend including additional tests to strengthen the evaluation.

**Questions To Address In The Rebuttal:**

What is the performance on a larger dataset? Is there any failure cases? What are the limitations of the proposed method?

**Special Issue:**

No

---

> ### Author Response · Authors · 2025-03-07
>
> We thank the reviewer for their comments and interest in the method.
>
> Q: What is the performance on a larger dataset? More quantitative or qualitative comparisons?
>
> A: As mentioned also to reviewer 3, we agree that larger-scale experiments should be addressed in future work. One initial reasoning to focus on smaller 2D models was the possibility to leverage pre-trained transformers and evaluate their transferability. However, we also trained large 3D VQ diffusion models from scratch and experimented with latent space dimensions of up to 16x24x20 - hence a  sequence length of 7’680 (and 59 million binary attention scores for each sample) - with success.
> To test the influence of the reduction in channel size for the value tensor, we also trained models without this step and achieved 3.9% Dice improvements for 2D segmentations on average.
>
>
> Q: Figure 2 is unclear and does not effectively convey the main point; the logical flow could be improved.
>
> A: This is a very good point. We have accordingly recreated Figure 2 to better convey the concept of our approach and uploaded the revised version under rebuttal above.
>
>
> Q:  Is there any failure cases? What are the limitations of the proposed method?
>
> A: So far we have limited ourselves to single-scale (vanilla) transformer models similar to the original ViT. This may lead to inaccuracies of finer anatomical details. Hence future work should consider multi-scale feature hierarchies (in 3D) to better evaluate long-range attention across different resolution levels.

---

### Official Review · Reviewer_inhj · 2025-02-22

**Confidence:** 2
**Preliminary Rating:** 4
**Recommendation:** Poster

**Summary:**

This paper proposed a method that aimed to address the high computation burden in attention mechanism. The author combined bitwise Hamming distance with learnable scalar query and key weighting which improves resource efficiency by 16-32× in arithmetic operations and memory bandwidth as the author claimed in theory. The author evaluates this method on three tasks—image classification, 2D MRI segmentation, and 3D high-resolution diffusion models and it shows competitive performance compared with baseline and other alternatives.

**Strengths:**

1. This paper proposed a "new derivation of diﬀerentiable binarised matrix-multiplication through the concept of bilinear interpolation of Hamming distances and Bernoulli sampling" and a trainable unary weighting per token to address the computational burden.
2. The author demonstrated their method on three tasks and showed improvements compared with the baseline.
3. The code is given.

**Weaknesses:**

1. Figure 1 is not clear. It would be better if the author can re-draw the figure and make it clear.
2. No comparisons with full attention block. I'd like to see the degradation/improvements compared with the full size attention block.

**Detailed Comments:**

As stated in the weakness part, figure 1 is not clear. If possible, add more experiments compared with the full size attention block.

**Justification Of The Preliminary Rating:**

I'm not very familiar and confident in the theory part of this paper. I'm not sure if there's any existing work addressing the computation burden in attention mechanism in this way or a similar way. Thus, the novelty for this method is unsure for me. Otherwise I  think this paper is valuable.

**Questions To Address In The Rebuttal:**

1. Can the author re-draw figure 1 to make it more informative and clear?
2. Add comparisons with full size/ original attention model (maybe a smaller architecture)?

---

> ### Author Response · Authors · 2025-03-07
>
> We thank reviewer for their insightful comments and appreciation of our approach.
>
> Q: The concept figure is not clear and should be redrawn.
>
> A: As also suggested by the second reviewer we have recreated the concept figure to make it more informative.
>
> Q: Add comparisons with full size/ original attention model
>
> A: Regarding this second relevant point asked for the rebuttal - we think this could be a slight misunderstanding. The baseline / full precision experiments are in fact the full attention computation. Possibly the reviewer was referring to the other approximation, i.e. the channel reduction for the value tensor. Following this suggestion we re-trained the models without this modification for the 2D segmentation tasks. We found that a modest gain of on average 3.9% can be reached across all tests when using the full sized value tensors.

---

### Author Rebuttal · Authors · 2025-03-07

**Rebuttal:**

revised manuscript for rebuttal

**Supporting Material:**

/attachment/34a761cc119b37e47ae5df30083d5c0afe5c66cc.pdf

---

> ### Comment · Reviewer_5rRZ · 2025-03-11
>
> In the revised version, the image of figure 6 seemed to be missing during the compilation of the latex file. Please make sure to include it in the final version.

---

> > ### Author Response · Authors · 2025-03-13
> >
> > Thanks a lot for noticing this. Yes I will reinsert this figure in the final version.

---

### Meta-Review · Area_Chair_G3pz · 2025-03-18

**Recommendation:** Accept (Poster)
**Confidence:** 5

**Metareview:**

The authors have adequately addressed the concerns raised by the reviewers during the rebuttal period. From the reviewers' perspective, the proposed efficient attention module is valuable to the field for various applications.